# Monitoring and Surveillance of Patients with Gastroenteropancreatic Neuroendocrine Tumors Undergoing Radioligand Therapy

**DOI:** 10.3390/cancers15194836

**Published:** 2023-10-02

**Authors:** Thorvardur R. Halfdanarson, Nadine Mallak, Scott Paulson, Chandrikha Chandrasekharan, Mona Natwa, Ayse Tuba Kendi, Hagen F. Kennecke

**Affiliations:** 1Mayo Clinic, Rochester, MN 55905, USA; kendi.ayse@mayo.edu; 2Division of Molecular Imaging and Therapy, Oregon Health and Science University, Portland, OR 97239, USA; mallak@ohsu.edu; 3Texas Oncology, Dallas, TX 75230, USA; scott.paulson@usoncology.com; 4College of Medicine, University of Iowa Carver, Iowa City, IA 52242, USA; chandrikha-chandrasekharan@uiowa.edu; 5Langone Health, New York University, New York, NY 10016, USA; 6Providence Cancer Institute Franz Clinic, Portland, OR 97213, USA; hagen.kennecke@providence.org

**Keywords:** radioligand therapy, peptide receptor radionuclide therapy, neuroendocrine tumor, response, monitoring, surveillance, positron emission tomography, PET, imaging, somatostatin receptor

## Abstract

**Simple Summary:**

Radioligand therapy with [^177^Lu]Lu-DOTA-TATE is a therapeutic option for adult patients with somatostatin-receptor–positive gastroenteropancreatic neuroendocrine tumors (GEP-NETs). Patients undergoing radioligand therapy require diligent monitoring and surveillance. While published guidelines can provide guidance on general approaches to care, GEP-NETs are heterogeneous and the guidelines can be difficult to apply in individual and complex cases. In this article, we discuss emerging evidence on imaging, clinical biochemistry, and tumor assessment criteria in the management of patients receiving radioligand therapy for GEP-NETs as well as our own best practices. We offer practical guidance on how to effectively implement monitoring and surveillance measures to aid patient-tailored clinical decision-making.

**Abstract:**

Radioligand therapy (RLT) with [^177^Lu]Lu-DOTA-TATE is a standard of care for adult patients with somatostatin-receptor (SSTR)-positive gastroenteropancreatic neuroendocrine tumors (GEP-NETs). Taking advantage of this precision nuclear medicine approach requires diligent monitoring and surveillance, from the use of diagnostic SSTR-targeted radioligand imaging for the selection of patients through treatment and assessments of response. Published evidence-based guidelines assist the multidisciplinary healthcare team by providing acceptable approaches to care; however, the sheer heterogeneity of GEP-NETs can make these frameworks difficult to apply in individual clinical circumstances. There are also contradictions in the literature regarding the utility of novel approaches in monitoring and surveilling patients with GEP-NETs receiving RLT. This article discusses the emerging evidence on imaging, clinical biochemistry, and tumor assessment criteria in the management of patients receiving RLT for GEP-NETs; additionally, it documents our own best practices. This allows us to offer practical guidance on how to effectively implement monitoring and surveillance measures to aid patient-tailored clinical decision-making.

## 1. Introduction

Gastroenteropancreatic neuroendocrine tumors (GEP-NETs) are a group of cancers originating within the gastrointestinal tract and pancreas [1,2]. These cancers are biologically and clinically heterogeneous, ranging from well-differentiated, indolent tumors to high-grade, rapidly progressive tumors [1,3]. Despite this overall heterogeneity, most GEP-NETs express somatostatin receptors (SSTRs) on their cell surface [4]. Several therapeutic options are available for this complex disease, including surgery, loco-regional therapy, chemotherapy, somatostatin analogues, everolimus, sunitinib, and SSTR-targeted radioligand therapy (RLT) with [^177^Lu]Lu-DOTA-TATE (hereafter ^177^Lu-DOTATATE) [5]. Proactive disease monitoring and clinical surveillance are important for choosing among these therapies, assessing response and adverse events (AEs), and identifying opportunities to switch therapies in patients who have disease progression.

Disease monitoring strategies for patients with GEP-NETs undergoing RLT are distinct compared with other therapies because RLT relies on a unique precision nuclear medicine approach where SSTR expression is first confirmed and visualized using SSTR-targeted positron emission tomography (PET) imaging. Patients with SSTR-positive GEP-NETs are then considered potential candidates for SSTR-targeted RLT. In addition to patient selection, SSTR PET also has roles in patient follow-up and restaging of disease after completion of RLT [6]. Laboratory and safety evaluations should also be tailored to address the risk of certain radiation-associated AEs associated with RLT such as myelosuppression. Finally, patients receiving RLT are best monitored by a patient-centered multidisciplinary team (MDT) whose members together have the requisite expertise for RLT administration and the timely and accurate interpretation of all imaging, laboratory, clinical, and biomarker assessments.

Evidence-based guidelines provide key principles on the monitoring and surveillance of patients receiving RLT for GEP-NETs [7]. The straightforward application of these general principles for patient-tailored clinical decision-making is complicated by several factors. First, extensive heterogeneity in the biological behavior and clinical presentation of GEP-NETs necessitate more granular and fine-tuned patient monitoring strategies that cannot be encompassed by general guidelines [8]. Second, a lack of prospective clinical data forces clinicians to rely on institutional experience instead of level one evidence when choosing a specific monitoring strategy for their patients. This is especially problematic in the case of imaging, where clinicians must grapple with the multitude of anatomic and functional options available.

In this review, we use existing clinical data and our own experience to provide a clinical perspective on the roles of anatomic imaging, SSTR-targeted imaging, laboratory monitoring, circulating biomarkers, and clinical evaluations before, during, and after RLT for GEP-NETs. Additionally, we offer practical guidance on how to select and apply these surveillance methods in patient-tailored management.

## 2. Considerations and Practical Guidance for Patient Selection and Eligibility Determination for RLT

### 2.1. SSTR-Targeted Imaging

Patients must have SSTR-positive lesions on SSTR imaging to be considered eligible for SSTR-targeted RLT [9,10]. Lesion uptake higher than the liver, as indicated by a Krenning score of ≥3, is typically used to define SSTR positivity and eligibility for RLT. The Krenning score was originally based on [^111^ln]ln-pentetreotide scintigraphy (as used in the pivotal NETTER-1 trial of ^177^Lu-DOTATATE) [11], and a similar threshold-based approach has been adapted for SSTR PET [9,12,13]. As the Krenning scale is visually dependent on planar images and limited by semi-quantitative scoring metrics, efforts have been made to develop other imaging surveillance techniques with better sensitivity and specificity for tumor burden.

Several studies have investigated quantitative SSTR PET/computed tomography (CT) parameters (e.g., standardized uptake values [SUVs], tumor volume metrics, heterogeneity metrics, and tumor-to-liver ratios) to characterize SSTR expression, predict response to RLT, and potentially improve patient selection (Table 1) [14,15,16,17,18,19,20]. Most of these studies were conducted retrospectively in small patient populations, and the results among the studies have not always been concordant. Prospective validation of quantitative parameters is needed from large populations before their use in disease assessment, RLT patient selection, and prognosis can be recommended.

In terms of the specific imaging modality to assess SSTR status, current guidelines recommend the replacement of SSTR scintigraphy with the much more sensitive SSTR PET [6,21,22]. For institutions without a PET scanner, we recommend patient referral to a PET center instead of local scintigraphy imaging. There are three different radioligands currently approved and available for SSTR PET imaging, [^68^Ga]Ga-DOTA-TATE, [^64^Cu]Cu-DOTA-TATE, and [^68^Ga]Ga-DOTA-TOC. Current data [23,24,25] and our own experience do not indicate any clinically relevant differences in diagnostic accuracy among these agents that would lead to one being preferred over the others. The choice of a specific radioligand for imaging thus depends primarily on preference of the nuclear medicine team, availability, and cost considerations. To ensure consistency and reproducibility of imaging across the RLT pathway, it is advisable to use the same radioligand for pre- and post-RLT assessments.

Clinicians should be aware of the challenges and pitfalls associated with SSTR PET image interpretation. For example, accurate interpretation is confounded by incidental SSTR uptake due to factors not related to the presence of a GEP-NET, including non-tumor related pancreatic uptake (especially in the uncinate process), nonmalignant osteoblastic activity, inflammatory processes, benign masses such as uterine fibromas, and incidental meningiomas [26]. Additionally, some of the quantitative parameters, such as absolute SUVs, are dependent on the scanner, and cross-institution reproducibility is problematic. Consequently, we currently do not recommend using these parameters as a basis for clinical decisions [27].

Some of these challenges and pitfalls can be addressed by interpreting SSTR PET imaging in conjunction with contrast-enhanced anatomic imaging, as these modalities provide complementary information. For example, a common aim in SSTR PET image interpretation is the differentiation of physiologic versus GEP-NET-related radioisotope uptake in the pancreatic uncinate process, which may vary over time [28,29,30,31]. Correlation of SSTR PET with multiphasic CT or magnetic resonance imaging (MRI) may reduce the risk of false positives in this region. Pre-RLT imaging should thus incorporate SSTR PET along with multiphasic, contrast-enhanced anatomic imaging of the abdomen, including the liver and pancreas. Functional imaging with 2-[^18^F] fludeoxyglucose (FDG) PET also complements SSTR PET and has the potential to elucidate disease heterogeneity and improve patient selection for RLT [8,13]. This approach is especially useful in patients with grade 3 GEP-NETs or grade 1/2 GEP-NETs with mismatch lesions between SSTR PET and anatomic imaging. The NETPET scoring system, which incorporates information from both SSTR PET and 2-[^18^F]FDG PET (dual PET imaging), correlates with tumor grade and prognosis and has been validated as a prognostic biomarker in patients with GEP-NETs or bronchial NETs [32,33].
cancers-15-04836-t001_Table 1Table 1Select studies that have investigated the prognostic value of baseline SSTR imaging in patients with NETs receiving RLT.Study, ReferenceStudy DesignNo. and Type of PatientsType and Schedule of SSTR ImagingConclusionGabriel et al., 2009 [14]Single-center, prospective 46 patients with advanced NETs ^68^Ga-DOTATOCPET with CT or MRI before RLT and after the last therapy cycleTumor SUV on baseline ^68^Ga-DOTATOC PET was not predictive of response to RLTOksuz et al., 2014 [18]Retrospective40 patients with advanced, progressive NETs and evidence of SSTR expression^68^Ga-DOTATOC PET/CT 1–3 days before and 3 months after RLTPre-RLT SUV_max_ was prognostic for response to RLT but SUV_max_ threshold values predicting response are not yet knownKratochwil et al., 2015 [17]Single-center, retrospective30 patients with metastatic NETsBaseline ^68^Ga-DOTATOC PET/CT of liver metastasesSUV_max_ cutoff of >16.4 from ^68^Ga-DOTATOC PET/CT predicted response to RLTWetz et al., 2017 [20]Single-center, retrospective20 patients with progressive, metastatic NETs (grade 1 or 2) and SSTR-positive lesions^111^In-pentetreotide SPECT/CT before RLTSpatial heterogeneity of SSTR volume (asphericity) has potential in predicting response to RLTWerner et al., 2017 [19]Multi-center, retrospective142 patients who received RLT (77% GEP-NETs)^68^Ga-DOTATATE PET/CT or ^68^Ga-DOTATOC PET/CT before RLTBaseline image-based metrics of intratumoral SSTR heterogeneity correlated with PFS and/or OS; in contrast, baseline SUV_max_ and SUV_mean_ were not predictive of PFS or OS Graf et al., 2020 [16]Single-center, retrospective65 patients with progressive NETs (grade 1 or 2) referred for RLT^68^Ga-DOTATATE PET/CT or ^68^Ga-DOTATOC PET/CT before RLTHeterogeneity of SSTR expression (determined by visual assessment of PET/CT imaging) was associated with shorter OS and TTPOrtega et al., 2021 [34]Multi-center, prospective 91 patients with progressive NETs and adequate expression of SSTR2Baseline and interim ^68^Ga-DOTATATE PET/CT before second cycle of RLTBaseline quantitative imaging metrics of SSTR2 expression levels and heterogeneity were predictive of RLT response and PFSMetser et al., 2022 [35]Multi-center, retrospective 41 patients with progressive NETs and adequate SSTR2 expression^68^Ga-DOTATATE PET/CT at baseline There was no association between baseline levels of SSTR2 expression in a lesion and subsequent lesion response to RLT Durmo et al., 2022 [36]Single-center, retrospective 46 patients with unresectable, metastatic NETs and adequate SSTR2 expression^68^Ga-DOTATOC PET/CT at baseline and after 2 RLT cycles (interim) Baseline whole-body tumor volume was a negative predictor of RLT response and OS Zwirtz et al., 2022 [15]Single-center, retrospective34 patients with progressive grade 1 or NETs treated with RLT^68^Ga-DOTATATE PET/CT or ^68^Ga-DOTATOC PET/CT before RLT, after the 1st and 2nd cycles, and within 3 months of RLT completionA baseline metric incorporating Hounsfield Unit and SUV_mean_ was found to predict lesion progression after 3 RLT cycles^68^Ga = gallium-68; CT = computed tomography; MRI = magnetic resonance imaging; NET = neuroendocrine tumor; OS = overall survival; PET = positron emission tomography; PFS = progression-free survival; RLT = radioligand therapy; SPECT = single-photon emission computed tomography; SSTR = somatostatin receptor; SSTR2 = somatostatin receptor type 2; SUV_max_ = maximum standardized uptake value; SUV_mean_ = mean standardized uptake value; TTP = time to progression.

### 2.2. Risk of Adverse Events and Laboratory Monitoring

Myelosuppression is an important safety consideration for RLT, and an increased risk of AEs in patients with baseline renal impairment is possible [10]. Findings from a single-center, retrospective study suggest that baseline chronic kidney disease (CKD) is a risk factor for hematotoxicity while on RLT [37]. Patients with hepatic metastasis (mainly those with very high liver burden or impairment [38]) may have an elevated risk of hepatotoxicity after RLT [10], though RLT may continue to be a safe option for these patients [39]. Regardless of tumor burden, baseline hepatic dysfunction may predict premature discontinuation of RLT [40].

Accordingly, a battery of laboratory tests related to bone marrow, renal, and hepatic functions should be conducted at baseline [7]. The timing of baseline assessments varies in clinical practice. Joint guidelines from the North American Neuroendocrine Tumor Society and Society of Nuclear Medicine and Molecular Imaging (NANETS/SNMMI) recommend checking laboratory values approximately two weeks before the first RLT cycle, which coincides with when the treatment is typically ordered [7]. Some institutions may also review laboratory values shortly before RLT infusion (e.g., 48 h).

The NANETS/SNMMI guidelines also provide recommended minimum threshold index values for bone marrow, renal, and hepatic function before proceeding with SSTR-targeted RLT (Table 2) [7]. These thresholds serve as a good starting point, but similar to the timing of labs, these are likely to vary by institution and require adaptation to the individual patient.

## 3. Considerations and Practical Guidance for Monitoring during and after RLT

### 3.1. Safety Monitoring: Laboratory and Clinical Assessments

Patients undergoing RLT may experience a neuroendocrine hormonal crisis that occurs during or within days of the infusion [41]. The incidence of neuroendocrine hormonal crisis was <1% in the ERASMUS trial of ^177^Lu-DOTATATE in patients with SSTR-positive tumors (*n* = 811) [10], which is consistent with what we have observed in practice. Although rare, patients need to be monitored for signs and symptoms of neuroendocrine hormonal crisis while undergoing RLT (Table 2).

Laboratory assessments during and after RLT are important for early identification and management of post-treatment sequelae. During RLT, the NANETS/SNMMI guidelines recommend laboratory testing two weeks before each cycle [7]. Depending on the institution, blood counts are typically conducted 4–6 weeks after each cycle as this may capture the expected nadir of blood counts and inform decisions about subsequent treatment modifications. After the completion of RLT, the NANETS/SNMMI guidelines recommend clinical evaluation and laboratory tests 2–4 weeks and 2 (lab tests only), 3, 6, and 12 months after completion of RLT as a best-case scenario, assuming no AEs [7].
cancers-15-04836-t002_Table 2Table 2Safety and monitoring considerations for ^177^Lu-DOTATATE RLT in GEP-NETs.Safety ConsiderationsIncidence (%)Time CourseGeneral RLT Monitoring ConsiderationsIndividual Patient Considerations ^a^**Acute Reactions**



Neuroendocrine hormonal crisis (carcinoid crisis)ERASMUS [10]<1Systematic review [42]1–10Most commonly occurs at cycle 1 during or within a day of the infusion [10,42]Signs and symptoms of tumor-related hormonal release should be monitored (e.g., flushing, diarrhea, hypotension, and bronchoconstriction) [10]In cases of severe neuroendocrine hormonal crisis (carcinoid crisis), hospital admission for closer monitoring/management may be required**Adverse events during and after RLT**



Grade ≥3 myelosuppressionNETTER-1 [11]Transient in nature with resolution within 8 weeks for thrombocytopenia and neutropenia [7,11];In NETTER-1, median time to platelet nadir was 5.1 weeks after the first dose and median time to platelet recovery was 2 months [10]Baseline laboratory thresholds for RLT eligibility ^b^:
Hemoglobin >8 g/dL;White blood cell count >2000/mm^3^;Platelet count >70,000/mm^3^ (or 75,000/mm^3^ if using CTCAE grading criteria thresholds);
Blood cell counts should be monitored after each RLT cycle [10]. The timing of blood testing varies in real-world practice, but every 4–6 weeks is reasonable to capture the nadir of blood counts;It has been recommended that blood tests be performed 1, 3, 6, and 12 months after RLT completion and then at least yearly thereafter if results have been normal [7]The cause of cytopenia should be considered when assessing RLT eligibility. For example, if reduced platelet counts are due to splenic sequestration, the patient could still be a viable RLT candidate;Patients with abnormal blood tests should be monitored more closely (hematology consult, increased testing frequency) [7]  Thrombocytopenia 2 Anemia0 Lymphopenia 9 Leukopenia1 Neutropenia1Renal toxicity Grade ≥ 3 nephrotoxicity Grade ≥ 3 serum creatinine increaseNETTER-1 [11,43]51No therapy-related long-term renal failure in NETTER-1 [11,43] or ERASMUS [44];at 5-year follow-up of NETTER-1, mean change from baseline for CrCL was similar between RLT and control groups [43]Baseline laboratory thresholds for RLT eligibility: ^b^ eGFR < 50 mL/min/1.73 m^2^ not a contraindication [9]eGFR < 30 mL/min/1.73 m^2^ (use only in exceptional circumstances) [9]. Serum creatinine and CrCL should be monitored when patients are on RLT [10];It has been recommended that serum creatinine/eGFR be assessed at 1, 3, 6, and 12 months after RLT completion and then at least yearly thereafter if results have been normal [7]Risk factors for renal toxicity include hypertension, diabetes, and pre-existing RI [10,45];Patients with mild or moderate RI should have more frequent renal assessments [10]Hepatotoxicity  Hepatic tumor hemorrhage, edema, or necrosisERASMUS [44]<1 [10]ERASMUS [44]No therapy-related long-term hepatic failureLaboratory thresholds for RLT eligibility ^b^:Total bilirubin ≤3 × ULN;Serum albumin >3.0 g/dL
During RLT, transaminases, bilirubin, and serum albumin should be monitored [10];It has been recommended that liver panels be assessed at 1, 3, 6, and 12 months after RLT completion and then at least yearly thereafter if results have been normal [7]Anatomic imaging can help determine if elevated bilirubin is due to biliary obstruction rather than RLT-induced toxicity;Hepatotoxicity may be more common in patients with extensive hepatic metastases [46] or prior SIRT [47,48]**Long-term safety considerations**



MDS and leukemiaNETTER-1 [11,43]MDS: 1.8Leukemia: 0ERASMUS [44]MDS: 1.5Leukemia: 0.7Systematic review [49]RLT-related myeloid neoplasm: 2.61In NETTER-1, 2 cases of MDS occurred at 8 and 14 months after first RLT dose—no new cases of MDS or leukemia were reported during long-term follow-up [43];in ERASMUS, acute leukemia and MDS occurred after median follow-up durations of 55 and 28 months, respectively [44]No robust pre-emptive monitoring optionsRisk factors for MDS/leukemia include prior chemotherapy and radiotherapy (including SIRT) [50];Patients with persistent cytopenias merit closer monitoring (hematology consult; increased testing frequency) [7]^a^ In general, frequency of monitoring should be tailored to the risk of progressive disease (tumor grade and bulk), functional status, and concern for post-RLT adverse events [7]. ^b^ Laboratory thresholds are those recommended by the NANETS/SNMMI Procedure Standard for RLT with ^177^Lu-DOTATATE; these guidelines state that the thresholds should be considered as general RLT eligibility criteria [7]. ^177^Lu = lutetium-177; CrCL = creatinine clearance; CTCAE = Common Terminology Criteria for Adverse Events; eGFR = estimated glomerular filtration rate; MDS = myelodysplastic syndrome; NANETS = North American Neuroendocrine Tumor Society; RI = renal impairment; RLT = radioligand therapy; SIRT = selective internal radiation therapy; SNMMI = Society of Nuclear Medicine and Molecular Imaging; ULN = upper limit of normal.

### 3.2. Anatomic and Functional Imaging for Assessment of Response and Disease Progression

Precise monitoring and prediction of tumor response to SSTR-targeted RLT is one of the most challenging issues facing clinicians; the heterogeneity and variable progression patterns of GEP-NETs coupled with a lack of large-scale prospective and longitudinal imaging data are the main contributory factors [8]. Unfortunately, standardized response criteria based on anatomic imaging are not optimal for GEP-NETs, and there are no guidelines for the determination of response based on SSTR PET imaging. For these reasons, clinicians are guided by their institutional approach when choosing patient-tailored imaging strategies. From a practical standpoint, it makes sense for imaging scans to be obtained at a single institution and for the same protocols (especially in regard to contrast timing) to be used from diagnosis to follow-up in order to ensure consistency and reproducibility. Considerations for using imaging to assess response and disease progression are summarized in the subsequent sections and in Table 3.

#### 3.2.1. Anatomic Imaging

Anatomic imaging using CT and/or MRI continues to be a central and standard component of routine monitoring to assess response and identify disease progression post-RLT. This is reflected by its incorporation into clinical guidelines, including those specifically for RLT (NANETS/SNMMI) [7].

There are several considerations for the use of anatomic imaging to assess response and identify disease progression. Depending on the suspected site of disease, either CT or MRI may be preferred; CT is advantageous to evaluate the lungs and peritoneal disease, whereas MRI is superior for the evaluation of liver metastases. For example, MRI offers two very sensitive sequences for detection of hepatic metastases: diffusion-weighted imaging (particularly helpful for patients who cannot receive contrast agents) and delayed-phase imaging with a hepatobiliary contrast agent (the most sensitive MRI sequence to detect hepatic metastases) [51,52,53].

Multiphasic and contrast-enhanced techniques should be used for accurate detection of certain GEP-NETs (e.g., pancreatic and liver lesions) [8]. The arterial phase is important since most GEP-NETs (i.e., pancreatic primary or hepatic metastases) show arterial enhancement and are better detected with this phase. However, lesions are occasionally hypoenhancing and are better seen in subsequent phases. Most CTs currently conducted in real-world practice are single-phase and, in our experience, of limited sensitivity for hepatic metastases, underscoring the value of MRI in such situations.

Another consideration is the timing of serial anatomic imaging relative to RLT cycles. The NANETS/SNMMI guidelines recommend diagnostic imaging at 1–3, 6, and 12 months after the completion of all RLT cycles and then as clinically indicated [7]. This frequency can be modified based on individual patient and disease factors (e.g., tumor grade, tumor burden, patient preference, insurance coverage, etc.) [8]. There is also a question of whether anatomic imaging can be used during the RLT treatment period to inform changes in management. In the phase three NETTER-1 study, anatomic imaging was used to assess response to RLT midway through treatment (midpoint imaging) and after treatment completion [11]. It remains unclear from the limited data if early imaging adds value. For example, midpoint imaging (primarily with CT or MRI) rarely changed subsequent clinical management in a single-center retrospective study [54]. Midpoint imaging also has the potential to be confounded by transient, RLT-induced increases in lesion diameter. This phenomenon, known as pseudoprogression, is thought to be caused by radiogenic edema and inflammation [55]. Despite the uncertainty, midpoint imaging may prove beneficial for patients at high risk of progression who merit more extensive monitoring (e.g., patients with high-grade or previously clinically aggressive disease), and additional studies in this population may be warranted.

Finally, limitations of response criteria for the evaluation of response and disease progression in the GEP-NET setting require consideration. Tumor burden is typically assessed according to the Response Evaluation Criteria in Solid Tumors (RECIST). RECIST criteria are based on measurements of structural changes in tumor tissue using measurements obtained via CT/MRI. Use of RECIST in the GEP-NET setting has several well-established drawbacks (Table 3) [56] but remains the standard. Chief among these drawbacks is the heterogeneity in tumor biology observed at different anatomical sites, slow tumor growth (which may lead to the misclassification of progressive disease as stable disease), and lack of account for osseous lesions, as well as tissue characteristics such as tumor density and degree of enhancement. Given these limitations, alternative response criteria, such as Choi criteria, have been proposed. Choi criteria consider tumor density and require less change in tumor size to consider disease progression or response, and may be more appropriate for GEP-NETs than RECIST [15,57], but these criteria have not been validated for routine implementation in this setting.

#### 3.2.2. SSTR-Targeted Imaging

Clinical guidelines consider SSTR PET imaging appropriate for monitoring disease that cannot be reliably detected by anatomic imaging but can be readily seen on SSTR PET and for restaging patients after completion of SSTR-targeted RLT [6]. The rationale for these guidelines is that SSTR PET is more sensitive than anatomic imaging in certain scenarios (e.g., bone metastases) [6]. For example, in a retrospective study of RLT response evaluation methods, Huizing et al. found that at three months post RLT completion, 10% of patients had new lesions visible on SSTR PET/CT but not anatomic imaging [58]. A few studies have also found correlations between post-RLT SSTR PET imaging parameters and subsequent patient outcomes, but these metrics have not been prospectively validated (Table 4) [14,15,34,36,54,58,59,60,61,62].

Despite guideline support for the use of SSTR PET for monitoring disease predominantly seen on SSTR PET and for restaging after RLT completion, the optimal timing of SSTR-targeted imaging in relation to therapy is unclear. Practices range from 3–6 months to 9–12 months, and as with midpoint anatomic imaging, the value of midpoint SSTR PET imaging in predicting responses to RLT or other clinical outcomes remains not yet understood (Table 4) [14,15,34,36,54,58,59,60,61,62,63,64].

Some challenges and pitfalls have emerged from using SSTR PET for response assessment, in addition to those seen when using SSTR PET for patient selection (reviewed in Galgano et al.) [8]. First, changes in SSTR avidity may not necessarily be due to tumor growth or shrinkage but may instead be the result of other biological processes (e.g., de-differentiation) or the effects of other treatments (e.g., somatostatin analogs, chemotherapy) [8]. Second, changes in SSTR avidity may not be indicative of the overall response to treatment if there is considerable heterogeneity of SSTR expression within and across lesions [8,65]. Finally, there are neither standardized SSTR PET response criteria nor high-level prospective data to guide clinicians in the interpretation of SSTR PET imaging after RLT.

Clinical experience with RLT and SSTR PET has accumulated sufficiently to inform practical guidance and best practices. We agree with recent guidance from Galgano et al., which noted that disease progression should not be determined by minute changes in SUV on SSTR PET alone [8]. Clinicians should consider the patient’s clinical history and insights from complementary anatomic and functional imaging. 2-[^18^F]FDG PET/CT can be used to resolve discordant findings between morphologic imaging and SSTR imaging findings [46], particularly when high-grade or SSTR-negative disease is suspected. For patients receiving RLT, FDG-positivity has been shown to correlate with reduced overall survival, ref. [66] demonstrating the prognostic value of 2-[^18^F]FDG PET/CT imaging and its potential added utility to SSTR PET [32]. Until more robust data are obtained, and consensus is reached on optimal imaging and response assessment criteria, we recommend that the radiologic work-up strive to be reproducible and that caution be taken when attempting to interpret subtle changes in uptake on SSTR PET.cancers-15-04836-t004_Table 4Table 4Select studies that have investigated the utility of SSTR imaging protocols for response assessment and prediction of clinical outcomes in patients with NETs treated with RLT.Study, ReferenceStudy DesignNo. and Type of PatientsType and Schedule of SSTR ImagingResults**Interim SSTR Imaging (between the First and Last Cycles of RLT)**Haug et al., 2010 [59]Single-center 33 patients with well-differentiated metastatic NETs and eligible for RLTWhole-body ^68^Ga-DOTATATE PET/CT scans at baseline and 3 months after the first RLT cycleChanges in uptake on ^68^Ga-DOTATATE PET/CT predicted TTP and clinical improvement in symptomsMahajan et al., 2019 [61]Single-center, retrospective 16 patients with metastatic NETsPlanar whole-body scan for gamma emission 3 h post RLT injectionEarly post-RLT quality assurance scans were used to confirm successful administration of therapy and assess physiologic biodistribution of RLTOrtega et al., 2021 [34]Multi-center, prospective 91 patients with progressive NETs and adequate expression of SSTR2Baseline and interim ^68^Ga-DOTATATE PET/CT before second cycle of RLTInterim quantitative metrics of SSTR2 expression and tumor heterogeneity did not correlate with PFSNorman et al., 2021 [54]Single-center, retrospective 113 patients with advanced, progressive GEP-NETsMidpoint imaging before RLT cycles2, 3, or 4 with ^68^Ga-DOTATATE PET/MRI, FDG PET/MRI, MRI, or CTMidpoint imaging rarely changed subsequent clinical managementDurmo et al., 2022 [36]Single-center, retrospective 46 patients with unresectable, metastatic NETs and adequate SSTR2 expression^68^Ga-DOTATOC PET/CT at baseline and after 2 RLT cycles (interim) Change from baseline in semiquantitative and volumetric PET metrics had no association with RLT response or OSHeying et al., 2022 [64]Single-center, retrospective105 patients with SSTR-expressing NETs^68^Ga-DOTATATE PET/CT PET/MRI after 2 cycles of RLT Interim ^68^Ga-DOTATATE PET was more accurate than RECIST in assessing treatment response during RLT Zwirtz et al., 2022 [15]Single-center, retrospective34 patients with progressive grade 1 or 2 NETs treated with RLT^68^Ga-DOTATATE PET/CT or ^68^Ga-DOTATOC PET/CT before RLT, after the 1st and 2nd cycles, and within 3 months of RLT completionPatients showing ≥25% increase in the sum of SUV_max_ or ≥1 new lesion after 2 RLT cycles had worse OS**SSTR imaging after RLT completion**Gabriel et al., 2009 [14]Single-center, prospective 46 patients with advanced NETs and evidence of SSTR expression^68^Ga-DOTATOCPET with CT or MRI before RLT and after the last therapy cycleWhole-body ^68^Ga-DOTATOC PET at end of RLT was useful for early assessment of progressive diseaseKong et al., 2014 [60]Single-center, retrospective 68 patients with NETs and uncontrolled symptomatic disease or progressive disease^111^In-octreotide SPECT or ^68^Ga-octreotate PET imaging 3–6, 6–12, and >12 months after the last RLT cycle Patients with an SSTR imaging response (decrease in uptake relative to hepatic and splenic activity) had longer OS than those that did notHuizing et al., 2020 [60]Single-center, retrospective 44 patients with well-differentiated NETs and sufficient SSTR expressionCT/MRI, ^68^Ga-DOTATATE PET/CT, and serum CgA before, and 3 and 9 months after RLT^68^Ga-DOTATATE PET/CT detected new lesions compared with baseline earlier than anatomical imaging, but changes in quantitative ^68^Ga-DOTATATE uptake parameters after RLT were not associated with OSOpalinska et al., 2022 [62]Single-center, retrospective 12 patients with advanced NETs eligible for RLT^68^Ga-DOTATATE PET/CT at baseline and every 3–6 months after RLT completionChange from baseline in post-RLT corrected SUV_max_ metrics correlated with clinical response^68^Ga = gallium-68; ^111^In = indium-111; CT = computed tomography; CgA = chromogranin A; MRI = magnetic resonance imaging; NET = neuroendocrine tumor; OS = overall survival; PET = positron emission tomography; PFS = progression-free survival; RLT = radioligand therapy; SSTR = somatostatin receptor; SSTR2 = somatostatin receptor type 2; SUV_max_ = maximum standardized uptake value.

## 4. The Role of Circulating Biomarkers and Quality of Life/Patient Symptom Assessments

Clinicians frequently assess circulating biomarkers such as chromogranin A. GEP-NET guidelines and consensus statements do not recommend the general use of these biomarkers for clinical surveillance because of their limited accuracy; insufficient amount of quality data; lack of impact on management decisions; and/or absence of consensus on guideline panels (Table 3) [67,68,69]. We affirm these guidelines and do not recommend routine monitoring of nonspecific circulating biomarkers. Patients with functional pancreatic NETs have elevated plasma levels of functional neuropeptides (e.g., insulin, gastrin), and these specific biomarkers should be monitored before and after treatment to aid assessments of treatment response and disease progression [67]. Serial measurements of plasma or urine 5-HIAA may also be reasonable in patients with small intestinal NETs with carcinoid syndrome, but these probably do not need to be done more often than every 12 months.

There has been interest in the NETest and the peptide receptor radionuclide therapy predictive quotient (PPQ). The NETest is a multigene assay based on circulating NET-specific gene transcripts [70]. The PPQ is a prediction classifier that integrates circulating gene transcript signatures with tumor grade [71]. Clinical studies have demonstrated the potential of the PPQ to predict response to RLT and the NETest to serve as a surrogate biomarker of response to RLT [71,72,73]. Although these tests are promising, they remain investigational until prospectively validated as biomarkers in RLT clinical trials.

Analyses of clinical trial and real-world studies have shown that RLT provides quality of life benefits for patients with GEP-NETs [74,75,76] and improves symptoms associated with the disease [74,75,76,77,78]. Clinicians should track patient disease-related and carcinoid symptoms throughout the course of RLT to better evaluate responses and benefits from therapy and determine if any increase in symptoms is being caused by disease recurrence.

## 5. Conclusions

The practical guidance provided in this review is largely based on our individual and collective experiences with ^177^Lu-DOTATATE administered per the approved protocol [10] in patients with well-differentiated, G1 and G2 GEP-NETs. Due to the extensive heterogeneity of the disease, there is no “one size fits all” approach and plans must be customized for each individual patient.

The RLT treatment landscape for GEP-NETs is likely to evolve along several dimensions, including novel RLT-based combination therapies [79], targeted alpha-particle therapy [80], RLT retreatment [81], tailored RLT based on dosimetry [82], and use in patients with G3 disease (Table 5). If validated and established in the clinic, these RLT strategies will impact the monitoring and clinical surveillance of patients with GEP-NETs. For example, the type and frequency of laboratory monitoring may need to be adjusted based on the risk of AEs expected for a given combination therapy or therapeutic radioisotope.

Together with innovations in treatment, diagnostic advances in SSTR PET imaging interpretation and novel biomarkers for pre- and post-RLT disease assessments may also drive new opportunities for RLT patient monitoring. Research into quantitative SSTR uptake parameters is ongoing, with some promising results that require validation in prospective studies [62,83]. In the future, imaging may be supplemented by tracking specific biomarkers when appropriate.

## Figures and Tables

**Table 3 cancers-15-04836-t003:** Considerations for post-RLT disease progression and response monitoring [8,9].

Topic	Considerations	Key MDT Members for Addressing the Considerations
Consistency and reproducibility of imaging	Due to variability, perform initial and follow-up scans in the same imaging department/center For SSTR PET, standardized setup (e.g., choice of imaging radioligand) and workflow procedures along with careful data interpretation are important to enhance reproducibility over time	Radiologists, nuclear medicine specialists
Choice of imaging technique	Anatomic imaging remains a central component of patient monitoring Appropriate use criteria support use of SSTR PET for monitoring of NETs seen predominantly on SSTR PET and restaging after completion of RLT There is no consensus on which specific imaging modalities to use for an individual patient Suspected lesion site can help guide choice of imaging technique ○ Bowel, peritoneal disease: multiphase CT ○ Liver, pancreas: MRI (diffusion-weighted, hepatobiliary contrast in delayed phase) ○ Bone: SSTR PET Use of multiple imaging modalities provides complementary information and may highlight disease heterogeneity	Radiologists, nuclear medicine specialists medical oncologists
Timing of imaging assessments	Timing of pseudoprogression and utility of midpoint imaging Typical kinetics of response for RLT Influence of disease factors (burden, grade) on frequency of assessments Need to balance benefit of detecting early progression vs. costs/risk of frequent assessments	Radiologists, nuclear medicine specialists, medical oncologists, surgical oncologists
Criteria for response and disease progression based on imaging findings	Standard RECIST criteria for solid tumors have limitations when applied to NETs ○ Heterogeneity in tumor biology and response complicates the optimal choice and number of target lesions ○ Potential to misclassify PD as SD due to the slow growth rate of NETs ○ Targeted therapies may alter NET characteristics other than size (e.g., tumor density) ○ Potential confounding from pseudoprogression ○ Bone metastases are difficult to measure There are no standardized response criteria for SSTR PET	Medical oncologists, radiologists, nuclear medicine specialists
Circulating biomarkers	Lack of sensitivity, specificity, and predictive power Expert consensus is that current circulating biomarkers have limited clinical utility for most patients (with the exception of functional pancreatic NETs)	Medical oncologists, endocrinologists
Quality of Life/symptoms	Subjective and nonspecific Changes in symptoms may reflect response to therapy or disease recurrence	Medical oncologists, gastroenterologists, endocrinologists
Clinical data for post-RLT monitoring/surveillance	Small number of studies which have not been validated	All members of the MDT

CT = computed tomography; MDT = multidisciplinary team; MRI = magnetic resonance imaging; NET = neuroendocrine tumor; PD = progressive disease; PET = positron emission tomography; RECIST = Response Evaluation Criteria in Solid Tumors; RLT = radioligand therapy; SD = stable disease; SSTR = somatostatin receptor.

**Table 5 cancers-15-04836-t005:** Future directions in RLT for GEP-NETs and their potential impact on patient monitoring and clinical surveillance.

Topic	Select Ongoing Studies or Results	Current Status and Potential Impact on Patient Monitoring
Novel RLT-based combination therapies	NCT04234568 (RLT + triapine) NCT04086485 (RLT + olaparib) NCT04750954 (RLT + peposertib) NCT05249114 (RLT + cabozantinib) NCT03044977 (2 RLTs together: ^131^I-MIBG and ^90^Y-DOTATOC)	Novel RLT-based combination therapies are investigational and have not been approved; If approved in the future, frequency and type of assessments may need to be adjusted based on specific safety considerations for each combination therapy
Targeted alpha-emitter therapy	NCT05153772 (^212^Pb-DOTAMTATE) Promising long-term results for ^225^Ac-DOTATATE [80]	Alpha-emitter therapies are investigational and have not been approved; If approved in the future, frequency and type of assessments may need to be adjusted based on specific safety considerations for each radioisotope
RLT retreatment	NCT04954820 (ReLUTH study) NET RETREAT study (SWOG/CCTG; RLT retreatment vs everolimus) NCT05477576 (RYZ101 [^225^Ac] vs investigator choice of everolimus, sunitinib, octreotide, or lanreotide) Systematic review/meta-analysis showed encouraging efficacy and safety [81]	RLT retreatment is investigational and has not yet been approved; If approved in the future, restaging with SSTR PET after initial RLT can inform eligibility and serve as a baseline disease assessment for RLT retreatment
Dosimetry-based RLT	NCT02754297 (dosimetry-based RLT) NCT04917484 (dosimetry- vs. standard-dose RLT)	Dosimetry-guided RLT is investigational and has not yet been approved
High-grade disease	NCT03972488 (NETTER-2 phase 3 study; RLT vs high-dose octreotide in grade 2 and 3 advanced GEP-NETs) NCT04919226 (COMPOSE phase 3 study; RLT vs SOC in grade 2 and 3 advanced GEP-NETs)	In the United States, ^177^Lu-DOTATATE is approved for the treatment of SSTR-positive GEP-NETs; Patients with high-grade disease may require more frequent monitoring
Multianalyte biomarkers	Circulating transcript assays have demonstrated promising results for monitoring response to RLT (NETest) and predicting response to RLT (PPQ) [73]	The NETest and PPQ assays are commercially available but have seen limited uptake; If validated and established in clinical practice, these biomarkers could improve patient selection, identify non-responders, and allow for early changes in treatment strategy

^131^I = iodine-131; ^177^Lu = lutetium-177; ^225^Ac = actinium-225; ^212^Pb = lead-212; ^90^Y = yttrium-90; CCTG = Canadian Cancer Trials Group; GEP-NETs = gastroenteropancreatic neuroendocrine tumors; MIBG = metaiodobenzylguanidine; PET = positron emission tomography; PPQ = PRRT predictive quotient; RLT = radioligand therapy; SOC = standard of care; SSTR = somatostatin receptor.

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
