# Peer review of "Monitoring and Surveillance of Patients with Gastroenteropancreatic Neuroendocrine Tumors Undergoing Radioligand Therapy"

_cancers, 2023, doi:10.3390/cancers15194836_

Round 1

Reviewer 1 Report

Thank you for an interesting and well-written manuscript on an important topic. It deserves publication, and I only have minor comments. 

1. Should other adverse effects be mentioned?

2. Could you describe shortly why Yt-DOTATATE is not included?

Reviewer 2 Report

The authors reviewed the clinical data of the radio ligand therapy of FDA approved 177Lu-DOTA-TATE, somatostatin II receptor targeted radiopharmaceutical for endocrine tumor therapy. This review comprehensively analyzed the massive clinical data of patients receiving 177Lu-DOTA-TATE radio ligand therapy with somatostatin-receptor–positive gastroenteropancreatic neuroendocrine tumors (GEP-NETs). Emerging evidence on imaging, clinical biochemistry, and tumor assessment criteria in the management of patients receiving radioligand therapy for GEP-NETs as well as their own best practices are extensively discussed. This review provides practical guidance for 177Lu-DOTA-TATE radio ligand therapy of GEP-NETs on how to effectively implement monitoring and surveillance measures to aid patient-tailored clinical decision-making. This review is attractive for GEP-NETs RLT clinicians and extraordinary helpful for future clinical practice. Therefore, I recommend this review to be published as is on Cancers.

Reviewer 3 Report

Simply a phenomenal review by Dr. Thor Halfdanarson and his team. Thorough, detailed, and meticulous review, with practical guidance and evident real-life experience and expertise. Great work that will be an amazing clinical tool for clinicians.  
